# A Multi-Experiment Investigation of the Effects Stance Width on the Biomechanics of the Barbell Squat

**DOI:** 10.3390/sports10090136

**Published:** 2022-09-14

**Authors:** Jonathan Sinclair, Paul John Taylor, Bryan Jones, Bobbie Butters, Ian Bentley, Christopher James Edmundson

**Affiliations:** 1Research Centre for Applied Sport, Physical Activity and Performance, School of Sport & Health Sciences, Faculty of Allied Health and Wellbeing, University of Central Lancashire, Preston PR1 2HE, UK; 2Faculty of Science & Technology, School of Psychology & Computer Sciences, University of Central Lancashire, Preston PR1 2HE, UK; 3Wigan Warriors RLFC, Wigan WN5 0UH, UK

**Keywords:** sport science, coaching, biomechanics, squat, muscle forces, kinetics

## Abstract

This two-experiment study aimed to explore habitual and manipulated stance widths on squat biomechanics. In experiment one, 70 lifters completed back squats at 70%, 1 repetition maximum (1RM), and were split into groups (NARROW < 1.06 * greater trochanter width (GTW), MID 1.06–1.18 * GTW and WIDE > 1.37 * GTW) according to their self-selected stance width. In experiment two, 20 lifters performed squats at 70%, 1RM, in three conditions (NARROW, MID and WIDE, 1.0, 1.25 and 1.5 * GTW). The three-dimensional kinematics were measured using a motion capture system, ground reaction forces (GRF) using a force platform, and the muscle forces using musculoskeletal modelling. In experiment two, the peak power was significantly greater in the NARROW condition, whereas both experiments showed the medial GRF impulse was significantly greater in the WIDE stance. Experiment two showed the NARROW condition significantly increased the quadriceps forces, whereas both experiments showed that the WIDE stance width significantly enhanced the posterior-chain muscle forces. The NARROW condition may improve the high mechanical power movement performance and promote the quadriceps muscle development. Greater stance widths may improve sprint and rapid change-of-direction performance and promote posterior-chain muscle hypertrophy. Whilst it appears that there is not an optimal stance width, these observations can be utilized by strength and conditioning practitioners seeking to maximize training adaptations.

## 1. Introduction

The back squat is a fundamental resistance/strength training exercise, and one of the most frequently utilized movements for the development of lower extremity strength and power [1]. As a multi-joint exercise, it is uniquely able to activate the quadriceps, hamstrings, gluteus, tibialis anterior, gastrocnemius, soleus and lumbar musculature [2] and, owing to being a closed chain kinetic exercise, it is also frequently exploited in rehabilitation environments [3]. Due to its popularity and structural association with performance in a plethora of athletic environments, the squat has garnered considerable research interest within the strength and conditioning literature. There are multiple variants of the squat, and a range of technique manipulations can be made in order to mediate different mechanical outcomes and training stimuli [4].

The commonly explored and adopted adaptations and manipulations include: the squat depth, stance width and foot placement angle [5]. The different types of executions may influence the biomechanics of the squat; thus, specific variations in the squat techniques may be adapted and manipulated in order to achieve more favorable training outcomes in distinct athletic groups. However, regarding the specific influence of the stance width, controversy exists regarding the employment of specific muscle groups, with anecdotal and lay texts supporting the notion that a narrow stance width increases the quadriceps muscle recruitment [6].

Specifically, the stance width has received relatively limited research attention, although recent publications in this area indicate that it is becoming a more prominent aspect of discussion in the strength and conditioning literature. Varying squat stance widths are utilized according to the athlete’s morphology and preferences [7], although the effects and efficacy of one stance over another is not well established. McCaw & Melrose [8] examined lower extremity muscle activation when performing the barbell back squat, with low and high barbell loads using stance widths of 75 and 140% shoulder distance. Their observations showed that the muscle activity in the gluteus maximus and adductor longus muscles was significantly larger in the wide stance condition. Escamilla et al. [5] examined three-dimensional joint kinetics and kinematics in a sample of national level powerlifters separated into narrow, medium and wide groups, based on their self-selected stance width. Their observations revealed that the hip was significantly more flexed, and the thigh was more horizontal, in the wide and medium condition compared to the narrow. In addition, the analysis of the joint moments showed that the knee extensor and ankle plantar flexor moments were significantly larger in the medium and wide conditions compared to the narrow. Escamilla et al. [7] investigated the effects of the stance width on the electromyographic activity of eight superficial thigh muscles during the leg press and barbell back squat exercises. Their findings show that the hamstring and gastrocnemius muscle activation was significantly greater in the wide squat compared to the narrow position. Similarly, Paoli et al. [9] examined the influence of the squat width on the lower extremity muscular activation, and their results indicate that the gluteus maximus activation was significantly greater in the wide squat condition. Sogabe et al. [10] explored the power production measured at the barbell using stance widths of 50, 100, 150 and 200% shoulder distance. Their findings show that peak power was significantly greater in the 150% shoulder distance, in relation to the other conditions. Finally, Lahti et al. [11] examined the kinematics, ground reaction forces (GRF) and sagittal/frontal plane hip and knee moments during wide (equal to 1.5 * greater trochanter width) and narrow (equal to 1.0 greater trochanter width) barbell back squats. The results show firstly that the knee flexion angle was statistically larger in the narrow stance width and that the hip-to-knee joint extension moment ratio was significantly greater in the wide stance. In addition, the angle of the ground reaction force vector relative to the horizontal axis was shown to be significantly reduced in the wide stance width condition.

### 1.1. Rationale

Despite the aforementioned scientific outputs concerning the effects of the stance width on the biomechanics of the barbell back squat, there has yet to be any scientific investigation that has concurrently examined the effects of the lifters’ habitually adopted stance width on the kinetics, three-dimensional kinematics and muscle forces, nor has there been any exploration of the effects of manipulating the stance width on these important parameters. Therefore, an exploration of the aforementioned areas may provide further insight regarding the effects of technique manipulations on the biomechanical outcomes during the barbell back squat that may be important for coaches and strength and conditioning practitioners seeking to promote distinct training stimuli.

### 1.2. Aims

As such, the aims of the current investigation are twofold. Firstly, experiment one aims, using a between-subjects’ design, to comparatively examine the effects of different self-selected stance widths on the kinetics, three-dimensional kinematics and muscle forces during the squat. Secondly, using a repeated measures study design, experiment two aims to explore the effects of manipulating the stance width on the same biomechanical parameters.

### 1.3. Hypotheses

The current investigation tests the hypotheses that both experiments will observe significantly lower resultant GRF vectors and statistically larger medially directed GRF’s and posterior chain muscle forces in the wide stance condition.

## 2. Materials and Methods

### 2.1. Ethical Approval

In accordance with the principles documented in the Declaration of Helsinki, the methods used in both experiments were approved by a University Science, Technology, Engineering Mathematics and Health (STEMH) ethics panel (Reference: 458/Date: 5/4/2016), and all participants provided informed consent.

### 2.2. Experiment 1

#### 2.2.1. Participants

Seventy male (age: 29.25 ± 5.40 years, stature: 177.25 ± 5.76 cm, mass: 81.14 ± 9.88 kg and 1 repetition maximum (1RM) back squat: 130.45 ± 22.79 kg) participants volunteered for this investigation. The participants were experienced in the high-bar back squat, with at least two years of experience and free from any lower extremity injury at the time of the data collection.

#### 2.2.2. Procedure

Three-dimensional kinematics were collected from each participant, using a motion capture system with eight cameras (Qualisys Medical AB, Goteburg, Sweden) and a capture rate of 250 Hz. To also collect the GRF data, piezoelectric force plates (Kistler, Kistler Instruments Ltd., Alton, Hampshire) were utilized, with a capture rate of 1000 Hz. The three-dimensional kinematics were synchronously collected using an analog-to-digital board.

Retroreflective markers were positioned using the set-up adopted by previous analyses to quantify the three-dimensional kinetics and kinematics of the back squat [12], and to allow the lower-extremity and trunk segments to be modelled in six degrees of freedom using the calibrated anatomical systems technique (CAST) [13], (Figure 1a). Two additional retroreflective markers were placed at either end of the bar to measure the bar kinematics during the squat. Static standing calibration trials were obtained from each participant prior to the commencement of the movement data collection.

#### 2.2.3. Squat Protocol

All the participants attended a familiarization session seven days prior to the data collection, in which they were shown and allowed to practice the experimental protocol and established their 1RM. The squat-testing protocol utilized in this study was adopted in accordance with a previously published protocol [14]. Briefly, all participants arrived at the laboratory two days after their last lower-body training session and completed a general warm-up, followed by progressive warm-up sets towards a target mass during the data collection of 70% 1RM. Five continuous, high-bar back squat repetitions using the participants’ self-selected squat technique/stance width were collected, using the same experimental footwear throughout (Adidas Powerlift, 3.0, Herzogenaurach, Germany). In accordance with the National Strength and Conditioning Association (NSCA) and UK Strength & Conditioning Association (UKSCA) guidelines, the participants descended in a controlled manner, kept both feet flat on the floor, preserved the proper breath control and maintained a constant/stable pattern of motion for each repetition. Each participant was observed by an NSCA-certified strength and conditioning specialist throughout.

#### 2.2.4. Processing

The markers were identified using the Qualisys Track Manager and exported in a Coordinate 3D (C3D) file format. All the data processing and extraction was undertaken using Visual3D (C-Motion Inc, Gaithersburg, MD, USA). As all participants were right-foot dominant and values were extracted from this side as symmetry was assumed [14], the bilateral side was utilized only in the calculation of the peak power at the center of the mass, for which the total vertical GRF applied to the body was required. The marker trajectories were filtered in accordance with the previous analyses using a Butterworth low-pass 4th order zero-lag filter, with a cut-off frequency of 6 Hz [15]. From each participant, the horizontal distance between the greater trochanter markers (taken from the static trial) and the center of mass of both foot segments were extracted. The quotient of the distance between the foot segments and the greater trochanter breadth was used to calculate the stance-width ratio [11]. The joint kinematics from the hip, knee, ankle and trunk were quantified using an XYZ cardan sequence, and the joint moments were calculated using inverse dynamics. All the data were normalized to 100% of the squat denoted by the first and second instances of the maximal hip flexion angles [15]. A further point at mid-lift was identified using the lowest vertical position of the bar [14].

The angle at mid-lift and also the angular range of motion (ROM) from the initiation to the mid-lift were taken from each rotational axis from the hip, knee and ankle. In addition, the sagittal plane measures of flexion at the mid-lift and angular ROM were extracted from the trunk. The joint power in the sagittal plane of the hip, knee and ankle joints was calculated using the joint power function within the Visual3D. In accordance with Stone et al. [16], the integral of the power at each joint during the ascent phase was calculated, using a trapezoidal function to quantify the energy production at each joint. The percentage (%) of joint power contribution relative to the total power was calculated as the quotient of the energy production from each joint (described above) and the sum of the total energy production from the three joints [16].

The total squat duration (s) was obtained, using the overall time between the initiation to the end point of each repetition, and the duration of the ascent and descent phases (s) was also obtained. The percentage duration of the ascent and descent phases, expressed as a percentage (%) as a function of the total squat duration, was also taken. In addition, the maximum vertical velocity (m/s) and acceleration (m/s^2^) of the barbell during the ascent phase of the squat was quantified. The maximum extent to which the knee translated both anteriorly and laterally during the squat was calculated using the Visual3D. These net distances were normalized to the length of the shank and expressed as a percentage (%) [17].

The quadriceps, hamstring, gluteus maximum, gastrocnemius and soleus muscle forces were calculated using the musculoskeletal modelling approaches described within the scientific literature by Sinclair et al. [13]. The net muscle forces were normalized by dividing the net values by the body mass (N/kg), and the peak quadriceps, hamstring, gluteus maximus soleus and gastrocnemius forces, and the forces at mid-lift, were extracted. In addition, the impulse of these forces (N/kg·s) was calculated during the ascent and descent phases using a trapezoidal function. Finally, the rate of force development (RFD) at each of the aforementioned muscles during the ascent phase was also extracted by obtaining the maximum increase between the adjacent data points using a first derivative function within the Visual3D (N/kg/s).

From the force plate, the peak vertical GRF (N/kg) during the ascent phase was extracted. The RFD of the vertical GRF (N/kg/s) was also calculated by obtaining the peak increase in the vertical GRF force between the adjacent data points again, using a first derivative function within the Visual3D. The impulse of the vertical and medial GRF’s (N/kg·s) during the ascent and descent phases was obtained, again with a trapezoidal function. Furthermore, the peak power (W/kg) during the ascent phase was quantified as a product of the vertical GRF and the vertical velocity of the three-dimensional kinematic model center of mass within the Visual3D. In accordance with Lahti et al. [11], the angle of the resultant GRF vector, relative to the horizontal plane, was quantified at the instance of mid-lift by taking the product of an inverse tangent function and the quotient of the medial–lateral and vertical GRFs.

#### 2.2.5. Statistical Analyses

For comparative analyses, the participants were split according to their stance-width ratio, which was split according to the 33.3 and 66.6 percentiles, allowing the creation of three separate groups: NARROW (1.06 ± 0.08 * greater trochanter width, N = 22); MID (1.18 ± 0.02 * greater trochanter width, N = 17); and WIDE (1.37 ± 0.12 * greater trochanter width, N = 31). All experimental variables are presented as the mean and standard deviations (SD) for each of the three stance-width groups. Differences between the three groups were examined using between-subjects linear mixed effects models with the stance-width group modelled as a fixed factor and random intercepts by the participants [18]. For the linear mixed models, the mean difference (*β*), t-value (t) and 95% confidence intervals (95% C.I.) of the difference are presented. All the analyses were conducted using the Statistical Package for the Social Sciences (SPSS) v27 (IBM, SPSS, New York, NY, USA), and the statistical significance was accepted at the *p* ≤ 0.05 level. In the interests of conciseness and clarity, only the experimental variables that presented with statistical significance are presented in the results section.

### 2.3. Experiment Two

#### 2.3.1. Participants

Twenty male (age: 32.10 ± 10.02 years, stature: 182.10 ± 4.88 cm, mass: 89.95 ± 13.21 kg and 1RM back squat: 130.55 ± 21.79.72 kg) participants took part in experiment two. The same inclusion criteria as experiment one were adopted.

#### 2.3.2. Procedure

The kinetic and kinematic information was obtained using the procedure and biomechanical modelling approach outlined in experiment one, and the participants once again wore the same footwear.

#### 2.3.3. Squat Protocol

The same number of repetitions, load, familiarization and warm-up procedure as in experiment one were adopted. However, as experiment two examined the effects of manipulating the stance width on the biomechanical outcomes, the participants were examined in three different conditions (NARROW, MID and WIDE). The NARROW condition was defined as being 1.0 * each participants’ greater trochanter width; the MID condition was 1.25 * greater trochanter width; and the WIDE was 1.5 * greater trochanter width [11]. The order that the participants performed in these three conditions was undertaken in a counterbalanced order, and the foot placement positions were marked clearly on the laboratory floor for each participant.

#### 2.3.4. Processing

The same processing techniques as in experiment one were adopted and the same experimental variables were extracted.

#### 2.3.5. Statistical Analyses

The differences between the three stance widths were examined using within-subjects linear mixed effects models with the stance-width condition modelled as a fixed factor and random intercepts by the participants. The same statistical principles and reporting as in experiment one were adhered to.

## 3. Results

### 3.1. Experiment 1

#### 3.1.1. Kinetic and Temporal Variables

The descriptive statistics for all the kinetic and temporal variables from experiment one that exhibited a statistical significance are presented in Table 1. The non-significant variables can be found in Appendix A.

The medial GRF impulse during the ascent phase was significantly larger in the WIDE, in relation to the MID (*β* = 0.57 _(95% C.I. = 0.19–0.94)_, t = 3.06, *p* = 0.002) and NARROW (*β* = 0.52 _(95% C.I. = 0.21–0.82)_, t = 3.36, *p* = 0.001), conditions (Table 1). In addition, the medial GRF impulse during the descent phase was significantly larger in the WIDE, in relation to the MID (*β* = 0.57 _(95% C.I. = 0.19–0.94)_, t = 2.00, *p* = 0.04) and NARROW (*β* = 0.39 _(95% C.I. = 0.04–0.79)_, t = 2.71, *p* = 0.006), conditions (Table 1). Finally, the angle of the GRF vector was significantly larger in the NARROW (*β* = 2.67 _(95% C.I. = 0.94–4.40)_, t = 3.10, *p* = 0.002) and MID (*β* = 2.99 _(95% C.I. = 0.89–5.09)_, t = 2.87, *p* = 0.003) conditions, compared to the WIDE stance width (Table 1).

The percentage of energy produced at the hip was significantly greater in the WIDE compared to the MID (*β* = 7.99, _(95% C.I. = 1.58–14.40)_, t = 2.51, *p* = 0.016) condition and, conversely, the energy produced at the knee was significantly greater in the NARROW, compared to the MID (*β* = 7.52, _(95% C.I. = 1.33–13.71)_, t = 2.45, *p* = 0.018), condition group (Table 1).

#### 3.1.2. Muscle Forces

The descriptive statistics for all the muscle force variables from experiment one that exhibited a statistical significance are presented in Table 2. The non-significant variables can be found in Appendix A.

The peak hamstring force was significantly greater in the WIDE condition (*β* = 16.67 _(95% C.I. = 2.29–31.04)_, t = 2.33, *p* = 0.02) compared to the NARROW (Table 2). Similarly, the hamstring force at mid-lift was significantly larger (*β* = 15.19 _(95% C.I. = 1.74–28.64)_, t = 2.26, *p* = 0.02) in the WIDE condition, compared to the NARROW (Table 3). The hamstring impulse during the ascent phase was significantly (*β* = 6.30 _(95% C.I. = 1.05–11.54)_, t = 2.41, *p* = 0.01) larger in the WIDE condition in relation to the NARROW (Table 4). In addition, the hamstring impulse during the descent phase was significantly (*β* = 6.86 _(95% C.I. = 1.91–11.80)_, t = 2.78, *p* = 0.005) greater in the WIDE condition in relation to the NARROW (Table 2).

The gluteus impulse during the descent phase was significantly (*β* = 2.51 _(95% C.I. = 0.37–4.66)_, t = 2.35, *p* = 0.02) larger in the WIDE condition in relation to the NARROW (Table 2). The soleus force at mid-lift was significantly (*β* = 2.74 _(95% C.I. = 1.33–5.35)_, t = 2.11, *p* = 0.04) larger in the WIDE condition in relation to the NARROW (Table 4). Finally, the soleus impulse during the ascent phase was significantly (*β* = 2.47 _(95% C.I. = 0.27–4.67)_, t = 2.25, *p* = 0.03) larger in the WIDE condition in relation to the NARROW (Table 2). In addition, the gastrocnemius force at mid-lift was significantly (*β* = 0.65 _(95% C.I. = 0.08–1.22)_, t = 6.25, *p* = 0.031) larger in the WIDE condition in relation to the NARROW (Table 3). Finally, the gastrocnemius impulse during the ascent phase was significantly (*β* = 9.76 _(95% C.I. = 6.23–13.29)_, t = 6.25, *p* < 0.001) larger in the WIDE condition in relation to the NARROW (Table 2).

#### 3.1.3. Kinematics

The descriptive statistics for all the kinematics from experiment one that exhibited a statistical significance are presented in Table 4. The non-significant variables can be found in Appendix A.

The hip abduction angle at mid-lift was significantly larger in the WIDE in relation to the MID (*β* = 4.25 _(95% C.I. = 0.03–8.64)_, t = 2.01, *p* = 0.04) and NARROW (*β* = 7.48 _(95% C.I. = 2.52–12.44)_, t = 3.07, *p* = 0.007) conditions (Table 3). In addition, the ankle eversion at mid-lift was significantly larger in the MID (*p* = 0.01) and NARROW (*p* = 0.008) conditions, compared to the WIDE (Table 4). Finally, the ankle rotation at mid-lift was significantly (*β* = 4.45 _(95% C.I. = 1.77–7.13)_, t = 3.33, *p* = 0.003) more externally rotated in the WIDE condition compared to the NARROW (Table 4).

### 3.2. Experiment 2

#### 3.2.1. Kinetic and Temporal Variables

The descriptive statistics for all the kinetic and temporal variables from experiment two that exhibited a statistical significance are presented in Table 3. The non-significant variables can be found in Appendix A.

The bar velocity was significantly greater in the NARROW compared to the MID (*β* = 0.06 _(95% C.I. = 0.03–0.09)_, t = 4.66, *p* = 0.002) and WIDE (*β* = 0.10 _(95% C.I. = 0.05–0.15)_, t = 4.88, *p* = 0.001) conditions. The angle of the GRF vector was larger in the NARROW condition in relation to the MID (*β* = 3.29 _(95% C.I. = 2.17–4.40)_, t = 6.64, *p* < 0.001) and WIDE (*β* = 6.31 _(95% C.I. = 4.89–7.74)_, t = 10.01, *p* < 0.001) conditions, and also in the MID condition (*β* = 3.03 _(95% C.I. = 2.11–3.94)_, t = 7.50, *p* < 0.001) compared to the WIDE (Table 4). The anterior knee displacement was greater in the NARROW (*β* = 4.42 _(95% C.I. = 1.06–7.78)_, t = 2.97, *p* = 0.02) and MID (*β* = 4.58 _(95% C.I. = 2.92–6.23)_, t = 6.27, *p* < 0.001) conditions compared to the WIDE (Table 3). The lateral knee displacement was greater in the NARROW compared to the MID (*β* = 2.80 _(95% C.I. = 0.83–4.77)_, t = 3.22, *p* = 0.011) and WIDE (*β* = 8.14 _(95% C.I. = 5.47–10.82)_, t = 6.89, *p* < 0.001) conditions, and in the MID compared to the WIDE (*β* = 5.34 _(95% C.I. = 2.88–7.81)_, t = 4.91, *p* = 0.001) (Table 4).

The total squat time was larger in the WIDE (*β* = 0.14 _(95% C.I. = 0.07–0.21)_, t = 4.36, *p* = 0.002) condition compared to the NARROW (Table 4). Also, the ascent time was larger in the WIDE compared to the NARROW (*β* = 0.09 _(95% C.I. = 0.05–0.12)_, t = 5.53, *p* < 0.001) and also the MID compared to the NARROW (*β* = 0.04 _(95% C.I. = 0.07–0.21)_, t = 3.16, *p* = 0.01) conditions (Table 4).

The peak power was larger in the NARROW compared to the MID (*β* = 1.45 _(95% C.I. = 0.86–2.48)_, t = 4.65, *p* = 0.001) and WIDE (*β* = 1.67 _(95% C.I. = 0.79–2.10)_, t = 5.04, *p* = 0.001) conditions. The peak vertical GRF was significantly larger in the NARROW compared to the WIDE (*β* = 0.56 _(95% C.I. = 0.25–0.87)_, t = 4.08, *p* = 0.003) condition (Table 3). The vertical GRF impulse during the ascent phase was larger in the WIDE condition compared to the NARROW (*β* = 0.54 _(95% C.I. = 0.23–0.86)_, t = 3.91, *p* = 0.004) (Table 3). The medial GRF impulse during the ascent phase was larger in the WIDE condition in relation to the MID (*β* = 0.59 _(95% C.I. = 0.47–0.72)_, t = 10.97, *p* < 0.001) and NARROW (*β* = 1.03 _(95% C.I. = 0.89–1.17)_, t = 16.36, *p* < 0.001) conditions, and also in the MID condition (*β* = 0.44 _(95% C.I. = 0.32–0.55)_, t = 8.40, *p* < 0.001) compared to the NARROW (Table 3). The medial GRF impulse during the descent phase was larger in the WIDE condition in relation to the MID (*β* = 0.67 _(95% C.I. = 0.47–0.87)_, t = 7.61, *p* < 0.001) and NARROW (*β* = 1.20 _(95% C.I. = 0.96–1.45)_, t = 11.04, *p* < 0.001) conditions, and also in the MID condition (*β* = 0.53 _(95% C.I. = 0.39–0.67)_, t = 8.79, *p* < 0.001) compared to the NARROW (Table 3).

The percentage of energy produced at the hip was significantly greater in the WIDE compared to the MID (*β* = 2.28, _(95% C.I. = 0.65–3.91)_, t = 3.16, *p* = 0.012) and NARROW (*β* = 3.71, _(95% C.I. = 1.31–6.11)_, t = 3.49, *p* = 0.007) conditions (Table 4). The percentage of energy produced at the knee was significantly greater in the NARROW compared to the MID (*β* = 1.98, _(95% C.I. = 0.23–2.56)_, t = 2.56, *p* = 0.031) and WIDE (*β* = 2.84, _(95% C.I. = 0.72–4.96)_, t = 3.03, *p* = 0.014) conditions (Table 3).

#### 3.2.2. Muscle Forces

The descriptive statistics for all the muscle force variables from experiment two that exhibited a statistical significance are presented in Table 5. The non-significant variables can be found in Appendix A.

The peak quadriceps force was larger in the NARROW condition in relation to the MID (*β* = 3.11 _(95% C.I. = 0.29–5.93)_, t = 2.50, *p* = 0.03) and WIDE (*β* = 6.61 _(95% C.I. = 3.59–9.62)_, t = 4.96, *p* = 0.001) conditions, and also in the MID condition (*β* = 3.49 _(95% C.I. = 1.78–5.21)_, t = 4.61, *p* = 0.001) compared to the WIDE (Table 5). In addition, the quadriceps force at mid-lift was larger in the NARROW (*β* = 8.70 _(95% C.I. = 5.35–12.05)_, t = 5.87, *p* < 0.001) and MID (*β* = 6.03 _(95% C.I. = 2.02–9.97)_, t = 3.45, *p* = 0.007) conditions compared to the WIDE (Table 5).

The peak gluteal force was larger in the WIDE condition in relation to the MID (*β* = 5.02 _(95% C.I. = 0.42–9.62)_, t = 2.47, *p* = 0.04) and NARROW (*β* = 5.86 _(95% C.I. = 0.54–11.17)_, t = 2.49, *p* = 0.03) conditions (Table 5). The gluteal force at mid-lift was larger in the WIDE condition in relation to the MID (*β* = 4.30 _(95% C.I. = 0.79–7.81)_, t = 2.77, *p* = 0.02) and NARROW (*β* = 5.93 _(95% C.I. = 0.37–11.49)_, t = 2.41, *p* = 0.04) conditions (Table 5). The gluteal impulse during the ascent phase was significantly larger in the WIDE condition in relation to the MID (*β* = 2.06 _(95% C.I. = 0.06–4.05)_, t = 2.33, *p* = 0.04) and NARROW (*β* = 2.74 _(95% C.I. = 1.00–4.48)_, t = 3.57, *p* = 0.006) (Table 5).

The peak hamstring force was larger in the WIDE condition in relation to the MID (*β* = 8.29 _(95% C.I. = 2.25–14.33)_, t = 3.10, *p* = 0.01) and NARROW (*β* = 9.48 _(95% C.I. = 2.69–16.27)_, t = 3.16, *p* = 0.01) conditions (Table 5). The hamstring force at mid-lift was larger in the WIDE condition in relation to the MID (*β* = 7.25 _(95% C.I. = 2.71–11.79)_, t = 3.61, *p* = 0.006) and NARROW (*β* = 9.90 _(95% C.I. = 2.30–17.49)_, t = 2.95, *p* = 0.02) conditions (Table 5). The hamstring impulse during the ascent phase was significantly larger in the WIDE condition in relation to the MID (*β* = 3.78 _(95% C.I. = 0.65–6.92)_, t = 2.73, *p* = 0.02) and NARROW (*β* = 5.14 _(95% C.I. = 2.60–7.68)_, t = 4.59, *p* = 0.001) (Table 5). The hamstring impulse during the descent phase was significantly larger in the WIDE condition in relation to the NARROW (*β* = 7.32 _(95% C.I. = 0.82–13.81)_, t = 2.55, *p* = 0.03) (Table 5).

The peak gastrocnemius force was larger in the MID condition in relation to the WIDE (*β* = 0.64 _(95% C.I. = 0.16–1.13)_, t = 3.00, *p* = 0.02) (Table 5). The gastrocnemius force at mid-lift was larger in the NARROW (*β* = 0.65 _(95% C.I. = 0.08–1.22)_, t = 2.56, *p* = 0.03) and MID (*β* = 0.78 _(95% C.I. = 0.37–1.19)_, t = 4.26, *p* = 0.002) conditions compared to the WIDE condition (Table 5)

The peak soleus force was larger in the MID condition in relation to the WIDE (*β* = 1.37 _(95% C.I. = 0.34–2.41)_, t = 3.01, *p* = 0.02) (Table 5). The soleus force at mid-lift was larger in the NARROW (*β* = 1.38 _(95% C.I. = 0.16–2.61)_, t = 2.54, *p* = 0.03) and MID (*β* = 1.66 _(95% C.I. = 0.78–2.54)_, t = 4.30, *p* = 0.002) conditions compared to the WIDE conditions (Table 5).

#### 3.2.3. Kinematics

The descriptive statistics for all the kinematic variables from experiment two that exhibited a statistical significance are presented in Table 6. The non-significant variables can be found in Appendix A.

The hip abduction angle at mid-lift was larger in the WIDE condition in relation to the MID (*β* = 2.20 _(95% C.I. = 0.96–3.44)_, t = 4.00, *p* = 0.003) and NARROW (*β* = 2.83 (95% C.I. = 0.76–4.91), t = 3.09, *p* = 0.01) conditions (Table 6). The hip abduction ROM was larger in the NARROW compared to the MID (*β* = 1.79 _(95% C.I. = 0.87–2.71)_, t = 4.41, *p* = 0.002) and WIDE (*β* = 3.86 _(95% C.I. = 2.38–5.34)_, t = 5.91, *p* < 0.001) conditions, and also in the MID compared to the WIDE (*β* = 2.07 _(95% C.I. = 0.83–3.31)_, t = 3.79, *p* = 0.004) condition (Table 6). The hip peak internal rotation was larger in the WIDE compared to the MID (*β* = 2.85 _(95% C.I. = 1.39–4.30)_, t = 4.43, *p* = 0.002) and NARROW (*β* = 5.50 _(95% C.I. = 3.06–7.93)_, t = 5.10, *p* = 0.001) conditions, and also in the MID compared to the NARROW (*β* = 2.64 _(95% C.I. = 1.31–3.99)_, t = 4.48, *p* = 0.002) condition (Table 6). The hip internal rotation ROM was larger in the WIDE compared to the MID (*β* = 4.23 _(95% C.I. = 2.20–6.27)_, t = 4.71, *p* = 0.001) and NARROW (*β* = 7.43 _(95% C.I. = 3.51–11.35)_, t = 4.29, *p* = 0.002) conditions, and also in the MID compared to the NARROW (*β* = 3.19 _(95% C.I. = 0.85–5.54)_, t = 3.09, *p* = 0.01) condition (Table 6).

The knee flexion at mid-lift was larger in the NARROW (*β* = 5.59 _(95% C.I. = 2.90–8.28)_, t = 4.70, *p* = 0.001) and MID (*β* = 4.14 _(95% C.I. = 2.36–5.92)_, t = 5.27, *p* = 0.001) conditions compared to the WIDE condition (Table 6). The knee flexion ROM was larger in the NARROW (*β* = 9.18 _(95% C.I. = 3.93–14.43)_, t = 3.96, *p* = 0.003) and MID (*β* = _7.95 (95% C.I. = 5.03–10.88)_, t = 6.15, *p* < 0.001) conditions compared to the WIDE condition (Table 6).

The dorsiflexion angle at mid-lift was significantly larger in the NARROW condition in relation to the MID (*β* = 2.35 _(95% C.I. = 0.84–3.86)_, t = 3.51, *p* = 0.007) and WIDE (*β* = 8.23 _(95% C.I. = 6.73–9.74)_, t = 12.41, *p* < 0.001) conditions, and also in the MID condition (*β* = 5.89 _(95% C.I. = 4.80–6.98)_, t = 12.25, *p* < 0.001) compared to the WIDE condition (Table 6). In addition, the ankle dorsiflexion ROM was significantly larger in the NARROW (*β* = 5.88 _(95% C.I. = 3.12–8.64)_, t = 4.81, *p*=0.001) and MID (*β* = 5.26 _(95% C.I. = 3.72–6.82)_, t = 7.68, *p* < 0.001) conditions compared to the WIDE condition (Table 6). The eversion angle at mid-lift was significantly larger in the NARROW condition compared to the WIDE (*β* = 5.69 _(95% C.I. = 0.99–10.39)_, t = 2.74, *p* = 0.02) condition (Table 6). The external rotation angle at mid-lift was significantly larger in the WIDE condition in relation to the MID (*β* = 3.13 _(95% C.I. = 0.94–5.33)_, t = 3.23, *p* = 0.01) and NARROW (*β* = 6.29 _(95% C.I. = 3.52–9.06)_, t = 5.14, *p* = 0.001) conditions, and also in the MID condition (*β* = 3.16 _(95% C.I. = 2.25–4.07)_, t = 7.88, *p* < 0.001) compared to the NARROW (Table 6).

## 4. Discussion

The aims of the current investigation were twofold. Experiment one comparatively examined the effects of different habitual stance widths on the kinetics, three-dimensional kinematics and muscle forces during the squat and experiment two explored the effects of manipulating the stance width on the same biomechanical parameters. To the best of the authors’ knowledge, this represents the first investigation to explore the aforementioned aims and may therefore provide further insight to coaches and strength and conditioning practitioners seeking to maximize training adaptations.

Importantly, the findings from experiment two revealed that the peak power output was significantly larger in the NARROW stance width condition in relation to the MID and WIDE conditions. This observation does not concur with that of Sogabe et al. [10], who found that the peak power output was significantly greater at the 150% shoulder distance. It is possible that the distinction between the studies relates to the approach that was adopted to quantify the peak power output during the squat, as Sogabe et al. [10] utilized an external dynamometer attached to the barbell. It is proposed that this observation is the cumulative product of the increased peak vertical GRFs and increased bar velocity that were also observed in the NARROW stance width condition. Notably, recent evidence has shown that, in order to optimally enhance power and performance in explosive athletic tasks, it is most effective to train with the resistance and exercise modality that maximizes the mechanical power output [19]. The findings from experiment two therefore suggest that the manipulation of an athlete’s set-up during the squat exercise to a NARROW stance width may improve the training stimuli necessary to mediate improvements in sports movements necessitating high mechanical power outputs.

In agreement with our hypotheses, the findings from experiments one and two show that both angle of the resultant GRF vector, as well as the impulse of the medial GRF in the ascent and descent phases of the squat, were significantly influenced as a function of the stance width. Specifically, the angle of the GRF vector was larger in the NARROW stance and the medial GRF impulse the greatest with a wider stance. These observations concur specifically with those of Lahti et al. [11], who also found increased medial GRFs and a reduced angle of the resultant force vector to the horizontal in the wide stance condition. It is proposed that these observations were mediated as a function simply of the different stance widths being examined across both experiments. As the resultant GRF vector initiates at the center of pressure and orients towards the body centee of mass, this finding relates to the altered position of the center of pressure relative to both the center of mass of the body in the WIDE stance, with conditions which served to enhance the magnitude of the medially directed GRFs. In relation to the distinct mechanical stimuli mediated through the increased medial GRFs, Lahti et al. [11] proposed that greater medial forces during the squat are able to improve proficiency in athletic disciplines and movements necessitating rapid changes of direction. Nagahara et al. [20] markedly showed that the impulse of the medial GRF was associated with enhanced sprint performance and plays a key role in enhancing propulsive performance. Therefore, this study indicates that individuals habitually adopting a WIDE stance width, and those manipulating their stance width to a wider position during the squat, may improve their performance in sprinting and rapid change-of-direction activities.

It is noteworthy that both experiments one and two also found the lower extremity muscle forces to be significantly influenced by the stance widths that were examined in this study. Specifically, it was revealed in experiment two that the NARROW squat condition mediated statistically increased quadriceps muscle forces, whereas, in support of our hypotheses, both experiments showed that the WIDE stance width condition was associated with significantly greater gluteus maximus, hamstring gastrocnemius and soleus forces. When also taking into account the observations from both experiments relating them to the joint energy production, this study indicates that the NARROW condition appears to arbitrate a knee-dominant squat strategy that targets the anterior chain musculature, whereas the WIDE condition produced a hip-dominant strategy [16] and targets the posterior chain musculature more effectively. This observation concurs with the electromyographic investigations of Escamilla et al. [7], McCaw & Melrose [8] and Paoli et al. [9], who found that wide-stance squat positions recruited the posterior chain musculature to a significantly greater extent. Furthermore, our observations in relation to the joint dominance are also supported by the examination of the joint moments by Lahti et al. [11], who showed that the hip-to-knee joint extension moment ratio was significantly greater when adopting a wide stance. Physiological analyses have demonstrated that the muscle forces are the primary mechanism responsible for the initiation of hypertrophic adaptations within the skeletal muscles [2], and also that the total muscle cross-sectional area is the central determiner of peak muscle force production [21]. Therefore, as the stimuli imposed by resistance training regulates the extent to which the skeletal muscles’ adaptive responses are made [22], this indicates that the utilization of a NARROW stance width may be advisable in athletes/coaches seeking to maximize quadriceps development, but that the utilization/manipulation of the stance width to a WIDE position appears to be the most effective mechanism to stimulate gluteus maximus, hamstring, gastrocnemius and soleus muscular development.

A limitation to the current study is that it represents an acute exploration of the effects of habitual and manipulated stance widths on the biomechanics of the squat. Whilst this investigation does represent an extension of the current literature base as concurrent kinetic, three-dimensional kinematic and muscle force indices are examined, it remains unknown as to whether alterations in stance width are able to mediate alterations in athletic performance indices. As such, a randomized and controlled intervention study is clearly warranted in order to explore the effects of manipulating the stance width of the squat on the performance-based outcomes in sport.

## 5. Conclusions

To conclude, although the influence of the stance width during the squat has been investigated previously, this study provides new information to the scientific literature by providing a two-experiment evaluation concerning the effects of habitual and manipulated stance widths on the kinetics, kinematics and muscle forces of the squat. Importantly, experiment two showed that the NARROW stance width significantly enhanced the peak power production during the squat and also enhanced the quadriceps forces. Both experiments also showed that the WIDE stance width significantly increased medial GRF impulses and mediated significant increases in the gluteus maximus, hamstring, gastrocnemius and soleus muscle kinetics. The NARROW condition may be able to mediate improvements in the high mechanical power movements and to promote the quadriceps muscle development. The increased medial GRFs suggest that greater stance widths may improve sprint and rapid change-of-direction performance and promote gluteus maximus, hamstring, gastrocnemius and soleus hypertrophy. As such, whilst it appears that there is not an optimal stance width, the observations from this study can be utilized by both strength and conditioning practitioners seeking to utilize and maximize training adaptations.

## Figures and Tables

**Figure 1 sports-10-00136-f001:**
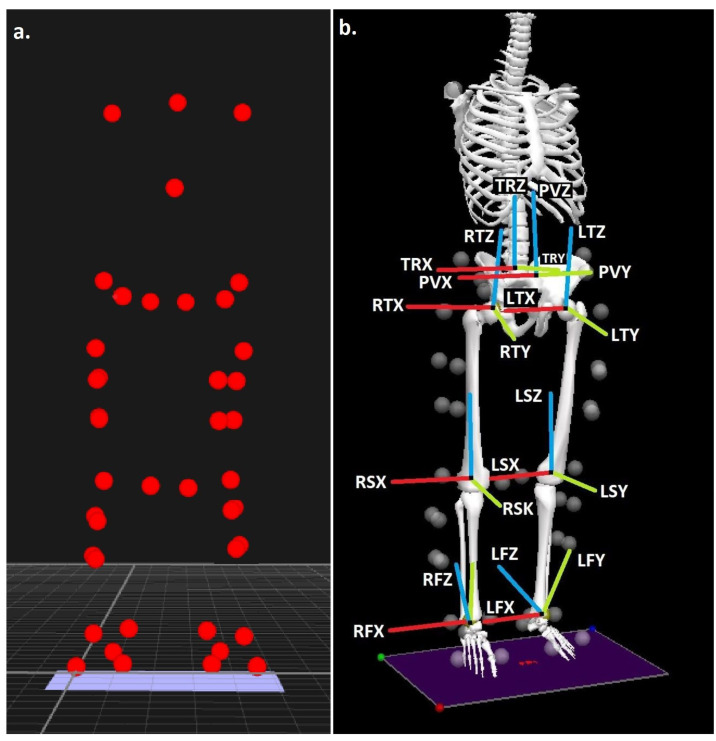
(**a**) Experimental marker locations and (**b**) trunk, pelvis, thigh, shank, and foot segments, with segment co-ordinate system axes (R = right and L = left), (TR = trunk, P = pelvis, T = thigh, S = shank, and F = foot), (X = sagittal, Y = coronal, and Z = transverse planes).

**Table 1 sports-10-00136-t001:** Kinetic and temporal parameters (mean ± SD) from experiment one as a function of each stance-width group.

	NARROW		MID		WIDE	
	Mean	*SD*	Mean	*SD*	Mean	*SD*
Medial GRF ascent impulse (N/kg·s)	1.35	0.45	1.30	0.38	1.87	0.62
Medial GRF descent impulse (N/kg·s)	1.28	0.48	1.32	0.44	1.72	0.65
Hip energy (%)	36.01	13.74	32.99	9.72	40.98	9.71
Knee energy (%)	54.13	11.40	58.99	10.50	51.47	8.93
GRF vector angle from the horizontal (°)	87.01	2.46	87.33	2.09	84.34	3.50
*Legend: GRF = ground reaction force*						

**Table 2 sports-10-00136-t002:** Muscle forces (mean ± SD) from experiment one as a function of each stance-width group.

	NARROW		MID		WIDE	
	Mean	*SD*	Mean	*SD*	Mean	*SD*
Gluteus descent impulse (N/kg·s)	8.52	3.02	9.61	2.26	11.03	4.35
Peak hamstring force (N/kg)	37.60	20.80	46.58	23.06	54.26	28.97
Hamstring ascent impulse (N/kg·s)	16.38	6.38	19.38	8.25	22.68	11.08
Hamstring descent impulse (N/kg·s)	15.39	5.93	18.05	6.35	22.25	10.49
Hamstring force at mid-lift (N/kg)	36.17	20.14	44.26	21.66	51.36	26.79
Gastrocnemius ascent impulse (N/kg·s)	5.89	2.07	4.88	1.83	4.75	1.67
Gastrocnemius force at mid-lift (N/kg)	6.64	2.37	5.74	2.13	5.36	2.10
Soleus ascent impulse (N/kg·s)	12.56	4.42	10.42	3.91	10.09	3.68
Soleus force at mid -ift (N/kg)	14.18	5.06	12.26	4.54	11.44	4.49

**Table 3 sports-10-00136-t003:** Kinetic and temporal parameters (mean ± SD) from experiment two as a function of each stance-width condition.

	NARROW		MID		WIDE	
	Mean	*SD*	Mean	*SD*	Mean	*SD*
Peak power (W/kg)	12.63	1.87	11.18	1.48	10.96	1.85
Peak bar velocity (m/s)	0.94	0.07	0.87	0.08	0.84	0.07
Total squat time (s)	2.05	0.21	2.11	0.20	2.19	0.22
Ascent duration (s)	1.02	0.09	1.06	0.11	1.10	0.10
Peak vertical GRF (N/kg)	11.72	1.20	11.47	1.39	11.16	1.06
Vertical GRF ascent impulse (N/kg·s)	8.55	1.65	8.77	1.86	9.10	1.80
Medial GRF ascent impulse (N/kg·s)	0.96	0.46	1.46	0.50	2.08	0.56
Medial GRF descent impulse (N/kg·s)	0.79	0.35	1.21	0.32	1.80	0.33
GRF vector angle from the horizontal (°)	88.58	3.25	85.30	3.71	82.27	3.84
Hip energy (%)	40.44	5.26	41.87	5.66	44.15	5.68
Knee energy (%)	51.17	4.46	49.19	4.20	48.33	4.38
Anterior knee displacement (%)	50.42	3.41	51.26	5.83	43.41	8.13
Lateral knee displacement (%)	26.30	11.15	23.50	10.43	18.16	11.96
*Legend: GRF = ground reaction force*						

**Table 4 sports-10-00136-t004:** Three-dimensional kinematics (mean ± SD) as a function of each stance-width group.

	NARROW		MID		WIDE	
	Mean	*SD*	Mean	*SD*	Mean	*SD*
Hip abduction at mid-lift (°)	−25.03	7.99	−32.52	4.64	−29.28	8.35
Ankle eversion at mid-lift (°)	−8.12	4.41	−8.77	5.85	−3.70	6.75
Ankle rotation at mid-lift (°)	−3.50	4.60	−1.06	6.47	0.95	5.02

**Table 5 sports-10-00136-t005:** Muscle forces (mean ± SD) from experiment two as a function of each stance-width condition.

	NARROW		MID		WIDE	
	Mean	*SD*	Mean	*SD*	Mean	*SD*
Peak quadriceps force (N/kg)	70.60	6.46	67.48	7.53	63.99	6.75
Quadriceps force at mid-lift (N/kg)	64.38	9.06	61.71	10.37	55.68	8.58
Peak gluteus force (N/kg)	28.13	10.46	28.97	11.23	33.99	17.46
Gluteus ascent impulse (N/kg·s)	14.07	4.19	14.76	3.96	16.81	6.39
Gluteus force at mid-lift (N/kg)	26.05	6.65	27.68	9.01	31.98	13.68
Peak hamstring force (N/kg)	60.42	18.38	61.61	18.81	69.90	26.56
Hamstring ascent impulse (N/kg·s)	30.04	8.21	31.40	7.55	35.18	11.18
Hamstring descent impulse (N/kg·s)	29.89	12.64	31.17	13.05	37.21	21.40
Hamstring force at mid-lift (N/kg)	56.35	11.50	59.00	14.63	66.25	20.08
Peak gastrocnemius force (N/kg)	5.78	1.04	5.90	1.16	5.25	1.23
Gastrocnemius force at mid-lift (N/kg)	4.50	1.42	4.63	1.94	3.86	1.80
Peak soleus force (N/kg)	12.34	2.23	12.59	2.47	11.21	2.63
Soleus force at mid-lift (N/kg)	9.62	3.02	9.89	4.13	8.23	3.85

**Table 6 sports-10-00136-t006:** Three-dimensional kinematics (mean ± SD) from experiment two as a function of each stance-width condition.

	NARROW	MID	WIDE
	Mean	*SD*	Mean	*SD*	Mean	*SD*
Hip abduction at mid-lift (°)	−20.33	8.24	−20.96	8.77	−23.17	9.38
Hip internal rotation at mid-lift (°)	0.21	9.52	2.85	9.18	5.70	8.86
Hip abduction ROM (°)	13.61	7.30	11.82	6.69	9.75	7.31
Hip internal rotation ROM (°)	15.09	8.71	18.29	10.13	22.52	11.45
Knee flexion at mid-lift (°)	134.13	4.44	132.68	5.29	128.54	6.92
Knee flexion ROM (°)	121.27	8.29	120.04	9.48	112.09	12.09
Ankle dorsiflexion at mid-lift (°)	27.68	6.20	25.33	7.11	19.44	6.99
Ankle eversion at mid-lift (°)	−7.15	7.51	−8.98	6.96	−12.84	2.16
Ankle rotation at mid-lift (°)	−3.81	3.90	−6.97	4.24	−10.10	6.99
Ankle dorsiflexion ROM (°)	30.54	3.10	29.93	3.76	24.66	5.11
*Legend: ROM = range of motion*

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
