# Peer review of "A Multi-Experiment Investigation of the Effects Stance Width on the Biomechanics of the Barbell Squat"

_sports, 2022, doi:10.3390/sports10090136_

Round 1
Reviewer 1 Report
The aim of this interesting two-experiment study was to explore the influence of different habitual and manipulated stance width on the kinetics, three-dimensional kinematics and muscle forces of the squat.
In the introduction, the authors introduce the reader sufficiently to the state of knowledge on this subject.
The research methods are adequately described and allow replication of these studies.
There are many parameters in the table. This may be incomprehensible to the potential reader. Are all parameters so important? I propose to show only the main ones.
The study adds to the current scientific knowledge, by providing a comprehensive two experiment evaluation concerning the effects of habitual and manipulated stance widths on the kinetics, kinematics and muscle forces of the squat. The manuscript is interesting and worth publishing.
Author Response
The aim of this interesting two-experiment study was to explore the influence of different habitual and manipulated stance width on the kinetics, three-dimensional kinematics and muscle forces of the squat.
In the introduction, the authors introduce the reader sufficiently to the state of knowledge on this subject.
RESPONSE: We thank you for your regarding our introduction section.
The research methods are adequately described and allow replication of these studies.
RESPONSE: We thank you for your compliment regarding our methodology section.
There are many parameters in the table. This may be incomprehensible to the potential reader. Are all parameters so important? I propose to show only the main ones.
RESPONSE: We have now presented only the statistically significant findings in the tables – the others will be presented as appendices.
The study adds to the current scientific knowledge, by providing a comprehensive two experiment evaluation concerning the effects of habitual and manipulated stance widths on the kinetics, kinematics and muscle forces of the squat. The manuscript is interesting and worth publishing.
RESPONSE: We thank you again.

Reviewer 2 Report
Dear authors,
The theme of the manuscript is very pertinent, as this two-experiment study was to explore the influence of different habitual and manipulated stance width on the kinetics, three-dimensional kinematics and muscle forces of the squat. However, some improvements should be considered to improve the manuscript.
Suggestions/comments are as follows:
- insert the study hypotheses in the introduction section;
- review the acromics throughout the manuscript, as the authors sometimes use the acromios before mentioning their meaning. To correct;
- Were there any familiarization sessions with the participants with the tests they were going to perform? If yes, put that information in the manuscript. If not, don't you think that this may have influenced the results obtained?
- put the "P" in lower case and in italics throughout the manuscript;
- the authors in the sub-section of participants must insert the manuscript was approved the ethics consent according to the declaration of helsinki;
- tables must have captions to make them easier to read;
- insert a section entitled "Discussion", since there is no separation between the results and the discussion;
- the discussion is very little developed. I suggest the authors discussed the results obtained with further studies published in the literature
Best regards.
Author Response
REVIEWER 2
Dear authors,
The theme of the manuscript is very pertinent, as this two-experiment study was to explore the influence of different habitual and manipulated stance width on the kinetics, three-dimensional kinematics and muscle forces of the squat. However, some improvements should be considered to improve the manuscript.
Suggestions/comments are as follows:
- insert the study hypotheses in the introduction section;
RESPONSE: Hypotheses have now been added to the paper.
- review the acromics throughout the manuscript, as the authors sometimes use the acromios before mentioning their meaning. To correct;
RESPONSE: This has now been completed throughout the paper.
- Were there any familiarization sessions with the participants with the tests they were going to perform? If yes, put that information in the manuscript. If not, don't you think that this may have influenced the results obtained?
RESPONSE: There were familiarization sessions included, this has now been added.
- put the "P" in lower case and in italics throughout the manuscript;
RESPONSE: This recommendation has now been incorporated into the paper.
- the authors in the sub-section of participants must insert the manuscript was approved the ethics consent according to the declaration of helsinki;
RESPONSE: This has now been included within the manuscript under section 2.1 ethical approval. We included a general section for this as at the beginning of the methodology, to avoid repeating ourselves for both experiments.
- tables must have captions to make them easier to read;
RESPONSE: Captions are now included for all tables.
- insert a section entitled "Discussion", since there is no separation between the results and the discussion;
RESPONSE: We thank you for seeing this, we have now corrected this with a formal discussion label which is denoted as section 4.
- the discussion is very little developed. I suggest the authors discussed the results obtained with further studies published in the literature
RESPONSE: We have now further developed the discussion section by making greater discussion of our observations against previous literature outlined in the introduction section, pertinent to this research question.
Best regards.

Round 2
Reviewer 2 Report
Dear authors,
First of all, I would like to thank you for the opportunity to review the manuscript again and to congratulate the authors for their improvement and quality. However, there are still some points that must be improved for the manuscript to present scientific quality to be published. Comments/suggestions are as follows:
- In the abstract section, authors must state the meaning of the acronyms NARROW, MID and WIDE.
- According to the journal's guidelines, the abstract should not contain more than 200 words and at this moment the abstract has 256.
- The tables still do not have legends about the meaning of the acronyms. Please post this information.
Best regards
Author Response
Dear authors,
First of all, I would like to thank you for the opportunity to review the manuscript again and to congratulate the authors for their improvement and quality. However, there are still some points that must be improved for the manuscript to present scientific quality to be published. Comments/suggestions are as follows:
- In the abstract section, authors must state the meaning of the acronyms NARROW, MID and WIDE.
RESPONSE: This has now been added to the abstract.
- According to the journal's guidelines, the abstract should not contain more than 200 words and at this moment the abstract has 256.
RESPONSE: The abstract is now reduced to be 200 words.
- The tables still do not have legends about the meaning of the acronyms. Please post this information.
RESPONSE: Legends for the tables have now been added for all of the acronyms.
Best regards